# Effectiveness of Tocilizumab in Patients with Severe or Critical Lung Involvement in COVID-19: A Retrospective Study

**DOI:** 10.3390/jcm11092286

**Published:** 2022-04-20

**Authors:** Daniel Chober, Bogusz Aksak-Wąs, Katarzyna Bobrek-Lesiakowska, Anna Budny-Finster, Ewa Hołda, Joanna Mieżyńska-Kurtycz, Grzegorz Jamro, Miłosz Parczewski

**Affiliations:** Department of Infectious, Tropical Diseases and Immune Deficiency, Pomeranian Medical University in Szczecin, 71-455 Szczecin, Poland; bogusz.aw@gmail.com (B.A.-W.); klesiakowska@gmail.com (K.B.-L.); budny.finster@gmail.com (A.B.-F.); ewaholda79@gmail.com (E.H.); jmiezynsk@gmail.com (J.M.-K.); grzjamro@gmail.com (G.J.)

**Keywords:** COVID-19, SARS-CoV-2, tocilizumab, mortality, dexamethasone, ICU, CRS

## Abstract

Introduction: Acute lung injury is associated with dysfunctional immune response to SARS-CoV-2. This leads to CRS, which require immunomodulatory treatments aiming to limit the excessive production of cytokines. The literature so far indicates the effectiveness of tocilizumab in patients with COVID-19-associated pneumonia, but there is no clear evidence of its effectiveness in patients with at least 50% lung involvement; therefore, we aimed to bridge this gap in knowledge. Materials and methods: Longitudinal data for 4287 patients with confirmed COVID-19 infection were collected between 1st March 2020 and 16th of January 2022. In total, 182 cases with lung involvement >50% and biochemical indicators of cytokine release storm (Il-6 >100 pg/mL) were selected and analyzed using non-parametric statistics and multivariate Cox models. Results: Among the 182 included patients, 100 (55%) were treated with TCZ, while 82 (45%) did not receive TCZ. The groups were balanced regarding demographics, lung involvement and biochemical markers. Overall mortality in the group was 63.1%. Mortality in the TCZ group was 58.0% compared to 69.5% (*n* = 57) in the non-TCZ group (*p* = 0.023). In multivariate Cox proportional hazards models, intravenous administration of tocilizumab was associated with lower probability of ICU admission (HR: 0333 (CI: 0.159–0.700, *p* = 0.004)) and lower mortality (HR: 0.57306 (CI: 0.354–0.927, *p* = 0.023)). Conclusions: Tocilizumab is effective as a treatment in the most severely ill patients, in whom the level of lung involvement by the inflammatory process can exceed 50% with coexisting biochemical indices of cytokine storm (Il-6 > 100 pg/mL).

## 1. Introduction

Since November 2019, when the first SARS-CoV-2 infection was identified, almost 430 million cases of this infection have been confirmed worldwide, with over 6 million deaths [1]. Disease severity ranges from asymptomatic or mild to severe, with symptoms of acute respiratory distress syndrome (ARDS) [2]. Severe and critical cases of COVID-19, which occur respectively in 14% and 5% of patients, may lead to acute respiratory distress syndrome (ARDS) due to COVID-19-associated pneumonia [3]. The COVID-19 mortality rates appears to be associated with the presence of severe ARDS and range widely, from 12 to 78 percent, with an average range of 25 to 50 percent [4].

Routinely used blood biomarkers for COVID-19 disease combined with radiologic assessment, including chest computed tomography (C-CT), may help to identify disease severity and mortality [5]. However, currently identified blood biomarkers such as interleukin-6 (IL-6), lactate dehydrogenase or procalcitonin, analyzed as standard of care on admission to hospital, have only moderate predictive value for mortality for COVID-19 compared to age or comorbidities [6,7]. The predictive value of imaging diagnostics based on computed tomography of the chest (C-CT) appears to be crucial in terms of severity and mortality [8]. The extent of lung involvement determined on the basis of the first C-CT examination correlates with clinical severity [9,10,11]. The French Society of Thoracic Imaging (SIT) recommends grading lung involvement as absent or minimal (<10%), moderate (10–25%), extensive (25–50%), severe (50–75%) or critical (>75%) [12].

COVID-19 treatment has evolved notably since the early pandemic period, with the current standard of care including antiviral agents such as remdesivir, molnupiravir and nirmatrelvir for the treatment of the early disease, supplemented with low-molecular-weight heparin, dexamethasone and supportive oxygen treatment [13]. Advanced disease stages with acute lung injury are commonly associated with dysfunctional immune response to SARS-CoV-2. This leads to cytokine release storm (CRS) with overproduction of interleukin-6 (IL-6), which requires immunomodulatory treatments aiming to limit the excessive production of cytokines [14]. To date, associations between increased concentration of cytokines such as Il-6, interleukin-8 (Il-8) and tumor necrosis factor alpha (TNF-α) and mortality have been identified [15,16]. Corticosteroids, anti-interleukin-1 receptor monoclonal antibody therapy (anakinra), anti-interleukin-6 receptor monoclonal antibody treatment (tocilizumab) and selective and reversible inhibitor of JAK1 and JAK2 kinases (baricitinib) have been included as a guideline-based standard of care in cases with CRS over time [13]. The administration of dexamethasone daily for up to 10 days reduced mortality among patients requiring oxygen support with or without mechanical ventilation [17]. Additionally, the anti-interleukin-6 receptor monoclonal antibody agent, tocilizumab (TCZ), has been reported to reduce mortality among hospitalized patients with COVID-19 pneumonia who did not require mechanical ventilation [18]. The use of tocilizumab, the IL-6 receptor blocker, is important as it may disrupt the inflammatory cascade at this key disease stage. However, some studies have not reported a benefit of TCZ use, mostly because of selection bias, failure to reach 80% power or due to very wide confidence intervals that lead to missing of potential benefit [19,20].

TCZ should only be used in cases with identified cytokine storm, with IL-6 levels exceeding 100 pg/mL [21] and features of COVID-19 associated pneumonia; however, data on its efficacy in the group of the patients with precisely identified severe disease are limited. To address this gap, we wished to evaluate the effect of TCZ treatment on mortality in the group of patients with severe (50–75%) and critical (>75%) lung involvement. A secondary goal of the study was to identify risk factors for ICU admission and mortality.

## 2. Materials and Methods

### 2.1. Study Groups

Our database contains information about patients hospitalized at the Regional Hospital in Szczecin, Poland, which was the largest COVID-19 treatment center in the West-Pomeranian region of Poland, where in total >13.000 disease cases were evaluated for admission during the current SARS-CoV-2 pandemics. Patients participating in the study were observed from 4th of March 2020 to 23rd of January 2022, when the database was closed. The database contains 4287 cases, all admitted to the hospital with COVID-19. Informed consent for participation in the analysis was obtained from all subjects, but the participation did not influence treatment options for the patients. All data were fully anonymized before statistical analyses.

In this study, we retrospectively analyzed the dataset of cases with severe or critical COVID-19 pneumonia based on chest computed tomography (C-CT). All patients presented with clinical symptoms of cough, dyspnea, fever (>38 °C) and oxygen saturation less than or equal to 90% prior to hospital admission. In every case, polymerase chain reaction (PCR) for severe acute respiratory syndrome coronavirus-2 (SARS-CoV-2) was performed using pharyngeal swabs, confirming infection with this virus, while pneumonia was confirmed using chest computed tomography(C-CT).

We assessed the lung involvement severity using the scale recommended by The French Society of Thoracic Imaging (SIT). The French Society of Thoracic Imaging (SIT) recommends grading lung involvement as absent or minimal (<10%), moderate (10–25%), extensive (25–50%), severe (50–75%) or critical (>75%).

All patients were admitted to the hospital on the basis of the assessment of the doctors on duty at the hospital emergency department. On admission to the hospital, the general condition, comorbidities and results of laboratory and imaging tests were taken into account. All laboratory and CT tests were performed immediately upon admission. Laboratory data used for this analysis were obtained within the first 24 h from admission. In cases where the tocilizumab was administered based on an increase in IL-6 in the subsequent days of hospitalization, the last IL-6 value prior to the drug administration was used.

All patients were treated in accordance with the current knowledge and guidelines of the Polish Association of Epidemiologists and Infectious Diseases Specialists (PTEiLChZ) [13,21]. If administered, the 5-day remdesivir course was started within 7 days of the onset of symptoms; if symptoms lasted longer, drug administration was not initiated. Remdesivir administered intravenously once daily for 5 days, with a loading dose of 200 mg on day 1, followed by maintenance doses of 100 mg. Chloroquine and lopinavir were not used in patients included for this analysis. Supportive treatment was applied to each of the patients. The supportive treatment included:-Antibiotic therapy (intravenous ceftriaxon was the drug of choice but could vary depending on the patient’s condition);-Oxygen therapy (low- or high-flow oxygen therapy or mechanical ventilation were used, while no extracorporeal ventilation was used among patients included in this analysis);-Intravenous rehydration based on individual needs;-Dexamethasone administered intravenously at a dose of at least 6 mg per day;-Low-molecular-weight heparin administered in prophylactic or therapeutic doses depending on patient condition.

This dataset was collected for all consecutive patients with severe pneumonia admitted to the Regional Hospital, Szczecin, Poland, between 4 March 2020 and 16 January 2022. The patients were divided into two groups for this analysis. In the first group, tocilizumab was administrated intravenously at a dose of 8 mg/kg (maximum dose: 800 mg) twice, 12 h apart. The second group was administrated only standard treatment in line with the guidelines. The control group consisted of patients who did not consent to the administration of tocilizumab and patients who, despite consenting, did not receive drugs for non-medical reasons (lack of drug availability in the hospital) or had medical contraindications for the drug use. Tocilizumab was administered according to guidelines, which did not change in this respect over time. Tocilizumab was administered only to patients with oxygen saturation (SpO_2_) <90% and interleukin-6 concentration >100 pg/mL. The final decision to use tocilizumab was at the discretion of the treating physician based on patient condition, consent for the drug use, indications listed above and infection risk. Before administering tocilizumab, acute CMV, EBV, HBV, HCV and HIV viral infections were ruled out, as well as acute toxoplasma gondii infestation based on serology.

As we wished to study the effects of TCZ in the group with severe pneumonia and CRS (indication for the TCZ use), the inclusion criteria for this study were as follows:-CT-confirmed COVID-19-associated severe (lung involvement 50–75%) or critical (lung involvement > 75%) pneumonia;-Biochemical indicators of cytokine release storm IL-6 level > 100 pg/mL (the level of interleukin-6 was established on the basis of the recommendations [21]);-Age > 18 years.

The endpoint for this analysis was death or discharge from the hospital.

### 2.2. Ethical Issues

The study protocol was approved by the Bioethical Committee of Pomeranian Medical University, Szczecin, Poland (approval number: KB-0012/92/2020). All patients gave their informed consent to participate in the study. Additional consent had to be given for the administration of off-label drugs, including tocilizumab. In every case, individual emergency consent was provided by the Bioethical Committee of Pomeranian Medical University, Szczecin, Poland, which was the standard procedure for the drug administration. The ethical committee worked on a daily basis from the beginning of the pandemic to provide consent for these cases. The study was conducted in accordance with the principles of the Declaration of Helsinki.

### 2.3. Sampling and Data Collection Methodology

In this study, we collected clinical data from medical records, including data on age, sex, comorbidities, treatment history, duration of in-hospital stay, duration of treatment in the ICU, survival statistics, baseline blood oxygenation levels, chest CT scan results and selected laboratory parameters (white blood cell count, neutrophiles count, lymphocytes count, hemoglobin levels, platelet count, procalcitonin levels, C-reactive protein levels, interleukin 6 levels, lactate dehydrogenase levels, d-dimer activity, bilirubin levels, troponin T levels, GGTP and aspartate and alanine aminotransferase activity levels). We created two groups based on procalcitonin concentration. The first group received a procalcitonin concentration below 2 ng/mL and the second group received a procalcitonin concentration above 2.0 ng/mL, as per laboratory sepsis prognostic recommendations. We divided creatinine concentrations into two thresholds, the first concentration below 2 mg/mL and the second above 2 mg/mL. This level simply corresponded to eGFR 30 mL/min/1.73 m^2^ for our group. The differences in the concentration of troponins T were most likely due to the secondary origin of renal failure.

### 2.4. Statistics

Clinical and baseline laboratory characteristics were calculated for non-parametric statistics, as data did not follow the normal distribution patterns. Statistical comparisons were performed using Mann–Whitney U tests for non-parametric statistics. Multiway tables were made with consideration of chi-square, Pearson and NW results. Confidence intervals (CIs) and interquartile ranges (IQRs) are indicated where appropriate.

A Kaplan–Meyer cumulative mortality test was calculated with the statistical significance of survival data analyzed using the log-rank test. Unadjusted and multivariate Cox proportional hazard models were used to assess the effects of the analyzed parameters on the risk of death and to calculate the hazard ratios (HR). For the final model, the best fit based on Akaike’s information criteria was selected. Here, *p*-values of 0.05 were considered significant.

Commercial software (Statistica 13.0 PL; Statasoft, Warsaw, Poland) was used for the statistical calculations.

## 3. Results

### 3.1. Clinical Characteristics of Patients with COVID-19

From our database, we selected cases with at least 50% lung involvement and concomitant CRS markers (IL-6 > 100 pg/mL) (Figure 1).

The dataset included 182 adult patients, with a median age 68.5 (IQR 61–76) years, with severe or lung involvement (Table 1). All cases were classified as having severe or critical lung involvement based on chest computed tomography (C-CT), with a median of lung involvement value of 59.61% (IQR 54.52–67.38). The key biochemical parameters indicative of advanced inflammation were significantly elevated. The median IL-6 was 177.5 pg/mL (IQR 129–287), with a concurrently low procalcitonin concentration (median 0.35 ng/mL (IQR 0.17–0.65)), clearly indicating a cytokine release storm. Attention is drawn to the extremely high median lactate dehydrogenase level (median 630 U/L (IQR 521.5–783.5)), which may indicate the risk of a severe course of the disease. Of 182 cases, all received glucocorticoids (dexamethasone administered intravenously at a dose of at least 6 mg per day), antibiotic therapy (drug of choice was ceftriaxone at at least 2 g daily but could vary depending on the patient’s condition), intravenous rehydration, supportive oxygen supplementary therapy and therapeutic low-molecular heparin. Of 182 cases, 57 (31%) received remdesivir. The patients’ chronic medications were maintained unless they required modification; for example, in diabetic patients, metformin was discontinued in all cases and insulin was used in treatment according to the glycemic profile.

The most common comorbidity was arterial hypertension, which was present in 63 (34%) cases. Diabetes was found in 38 cases (21%). In total, 25 (14%) cases were related with both diabetes and hypertension. Other comorbidities collected in our database occurred occasionally and had no statistical significance.

### 3.2. Clinical and Laboratory Data on TCZ Treated vs. Control Group

The study group included 100 (55%) patients who received TCZ and 82 (45%) controls treated who received standard care only. The median time from hospital admission to administration of the first dose of tocilizumab was 1 day (IQR 1–2 days).

There were no statistically significant differences between particular groups regarding age, comorbidities, percentage of lung involvement, white blood cell count, neutrophiles count, lymphocytes count, red blood cells count, hemoglobin level, hematocrit level, platelets count, CRP levels, interleukin-6 levels, lactate dehydrogenase levels, d-dimer activity, bilirubin levels, GGTP or aspartate and alanine aminotransferase activity levels. The groups differed statistically in terms of troponin T, procalcitonin and creatinine levels.

The median lengths of hospital stay were 15 (IQR 6–28) days, and16 (IQR 9.5–28.5) days for the TCZ group, and 11.5 (IQR 9.5–28.5) days for the non-TCZ group (*p* = 0.109).

### 3.3. Factors Associated with ICU Admission in the Analyzed Group

Out of 182 patients, 59 (32.4%) were admitted to ICU due to COVID-19-related pneumonia and its complications. Out of 59 patients admitted to ICU, 30 (51%) were treated with TCZ and 29 (49%) were treated with standard care only. Of 123 patients not admitted to ICU, 52 (42%) were treated with TCZ and 71 (58%) were treated with standard care only (*p* = 0.27). All patients in ICU were on invasive mechanical ventilation. We analyzed the group in terms of statistical significance of risk factors for admission to ICU. Statistically significant factors in our analysis were age, red blood count, CRP levels, D-dimers and troponin T levels. The median age was higher in groups admitted to ICU. The median percentage of lung involvement and median red blood cell count were higher in groups not admitted to ICU.

Comorbidities were important factors in admission to the ICU. There were statistically significant differences between the groups regarding frequency of hypertension and diabetes (Chi2 *p* = 0.001), which affected the risk of admission to the ICU. Out of 63 cases related to hypertension, 38 were admitted to ICU (60%). Out of 38 cases related to diabetes, 26 were admitted to ICU (68%). Additionally, there were notable differences between the groups regarding remdesivir use (Chi2 *p* = 0.001). Out of 57 cases treated with this agent, 28 were admitted to ICU (49%). All statistically significant factors are presented in Table 2.

We created multivariate Cox proportional hazards models for all statistically significant factors and for the administration of tocilizumab as the main goal of our study. In the multivariate Cox proportional hazards models (Table 2), higher CRP levels (HR: 1005 (CI: 1001–1008), *p* = 0.004) and diabetes as a concomitant disease (HR: 2116 (CI: 1119–4002), *p* = 0.02) were associated with higher probability of ICU admission. Administration of tocilizumab was associated with lower probability of ICU admission, but it was not statistically significant (HR: 0.574 (CI: 0.320–1031), *p* = 0.063) (Table 2).

### 3.4. Overall Mortality Risk

Overall mortality in the group was high. Out of 182 patients, 115 (63.1%) died from COVID-19-related pneumonia and its complications, while 67 (36.9%) survived. Out of 59 patients admitted to ICU, 45 (76.2%) died and 14 (23.8%) survived. The median length of treatment for the group who did not survive was 9 (IQR 3.5; 19) days. The median length of treatment for the group who survived was 27.5 (IQR 15; 40) days. 

We also analyzed the group within the context of death risk factors, regardless of tocilizumab use (Table 3). Higher mortality factors in our analysis were associated with age, gender, lymphocyte count, red blood cell count, hemoglobin level, hematocrit, procalcitonin level, creatinine level, troponin T level and the activity levels of aminotransferases and GGTP. The concentrations of interleukin-6 and the activity levels of lactate dehydrogenase did not differ significantly.

Diabetes was associated with higher mortality risk (10 percent points), but there were no statistically significant differences (Chi2 *p* = 0.25). Additionally, hypertension and remdesivir use were not associated with mortality risk differences. All statistically significant factors are presented in Table 3.

### 3.5. Tocilizumab Associated Mortality

The mortality rate in the TCZ group was 58.0% (*n* = 58) compared to 69.5% (*n* = 57) in the non-TCZ group (*p* = 0.02341). The median length of hospital stay for the TCZ group was 16 (IQR 9.5; 28.5) days, while it was 11.5 (IQR 3; 24) days for the non-TCZ group. For those who survived, the median lengths of treatment were 28 (IQR 15; 41) days for the TCZ group and 22 (IQR 14; 35) days for the non-TCZ group. For those who died, the median lengths of treatment were 12 (IQR 6; 20) days for the TCZ group and 6 (IQR 2; 15) days for the non-TCZ group (Figure 2).

We created multivariate Cox proportional hazards models for all statistically significant factors and for the administration of tocilizumab. As diabetes appears to be an important factor, we added it as an additional variable. In multivariate proportional Cox hazards models (Table 3), lower age, creatinine levels below 2 mg/dL and intravenous administration of tocilizumab were associated with lower mortality (HR: 0.466 (CI: 0.286–0.761), *p* = 0.003; HR: 0.499 (CI: 0.251–0.993), *p* = 0.003; and HR: 0.615 (CI: 0.394–0.956), *p* = 0.032, respectively). Higher troponin T levels were associated with higher mortality but the hazard ratio was low.

## 4. Discussion

The risk of death in cases with severe and critical lung involvement of COVID-19 is increased in patients with cytokine storm. In the literature to date, the influence of the cytokine storm and hyperinflammatory syndrome on the risk of death in the course of SARS-CoV-2 infection and the possibility of targeting them with immunomodulatory agents, especially corticosteroids and tocilizumab, have been well described [14,17,18,22]. However, the published data do not include any study that would directly observe the effectiveness of tocilizumab as an immunomodulating treatment in the most severely ill patients, in whom the level of lung involvement due to the inflammatory process would exceed 50% with coexisting biochemical indices of cytokine storm (Il-6 > 100 pg/mL). The WHO issued the first recommendation regarding IL-6 receptor blockers as the fifth version of the WHO living guidelines. The recommendations followed the publication of RECOVERY and REMAP-CAP trial publications. No changes have been made since then for the IL-6 receptor blocker recommendation [23].

In this study, we analyzed the effects of tocilizumab on mortality in patients with radiologically proven severe or critical COVID-19-related pneumonia with concomitant biochemical markers of cytokine storms. As the efficacy of tocilizumab was associated with the use of glucocorticoids, we would like to emphasize that all patients in this study received at least 6 mg of dexamethasone daily. With no statistically significant differences between treatment arms in terms of age, percentage of lung involvement, white blood cell count, neutrophiles count, lymphocytes count, red blood cells count, hemoglobin level, hematocrit, platelet count, CRP levels, interleukin-6 levels, lactate dehydrogenase levels, d-dimer activity, bilirubin levels, GGTP, aspartate and alanine aminotransferase activity levels and grouped levels of procalcitonin and creatinine, we found a significantly lower 60 day mortality rate in patients treated with tocilizumab in addition to standard treatment as compared to those treated with standard treatment only (dexamethasone, antibiotics, prophylactic or therapeutic low-molecular-weight heparin). The mean mortality in the group treated with tocilizumab was 58% compared to 69.5% in the group treated without this agent (Figure 2). The overall mortality in this study was high due to the eligibility criteria, which allowed only extremely severe patients to be included in the study. The low percentage of ICU admissions was associated with a constant lack of beds in the ICU, age-related disqualification and the phenomenon of “sudden death”, when the deterioration of vital functions occurs so quickly that it is impossible to transfer to the ICU. In a retrospective study of the predictors of mortality in patients with severe COVID-19 pneumonia, the overall mortality rate at 30 days was 56.60% [24]. The similar mortality rate suggests that just like the authors of the above-mentioned study, we qualified only the most seriously ill patients for our research.

Our results in reducing mortality are in line with the results of the randomized analysis of RECOVERY [25]. The analysis compared two groups of patients with regard to tocilizumab-related mortality. The mortality in the tocilizumab group was 31% out of 2022 patients compared to 35% of 2094 patients receiving standard care. The mortality differences in our work were significantly greater. This was due to differences in qualification for the study. Eligibility for randomization in the RECOVERY trial was based on the CRP level and dyspnea expressed as saturation <92%. Our analysis included only patients with severe dyspnea (saturation <90%), with at least 50% lung involvement by inflammation, and we used the concentration of interleukin-6 as the primary cytokine storm biomarker; in this group expected mortality was notably greater.

Another large study in patients with severe pneumonia was REMAP-CAP [26]. It was based on ICU patients who required respiratory or cardiovascular support. Patients were divided into a combined group treated with interleukin-6 receptor antagonists in addition to standard therapy and a group receiving only standard therapy. In-hospital mortality in the pooled interleukin-6 receptor antagonist group was 27% (108 out of 395 patients) compared with 36% (142 out of 397 patients) in the control group. The percentage decreases in mortality were similar, while the overall mortality in our study was diametrically greater. The main differences between our studies were based on the preliminary qualification of patients for the study, namely the definition of the disease severity. Severe COVID-19 was defined as receiving support for respiratory or cardiovascular failure in the ICU. The only biochemical markers analyzed were CRP and D-dimers. The severity of the pneumonia was not assessed radiographically.

The EMPACTA trial suggested that tocilizumab reduced the combined risk of mechanical ventilation and death among hospitalized patients with COVID-19 pneumonia who were not receiving mechanical ventilation, but the overall mortality was even higher in the tocilizumab group (10.4% (95%CI, 7.2 to 14.9) vs. 8.6% (95%CI, 4.9 to 14.7)) [18]. The differences between studies lay in the study qualification criteria. Eligibility for the EMPACTA study did not take into account the concentration of IL-6. In our case, the concentration of interleukin-6 was a crucial marker for TCZ implementation, in line with local guidelines (required a concentration >100 pg/mL). In the EMPACTA study, patients were excluded if they were receiving continuous positive airway pressure, bilevel positive airway pressure or mechanical ventilation. In our study, patients were eligible for tocilizumab administration regardless of the method of oxygen supplementation.

In a study by Stone et al., treatment with tocilizumab was considered ineffective in preventing intubation or death [20]. In this study, the inclusion criteria were based on the presence of at least two of the following: fever (body temperature > 38 °C) within 72 h prior to study entry, pulmonary infiltrates or the need for supplemental oxygen to maintain oxygen saturation above 92%. Additionally, it was required to meet one of the laboratory criteria: CRP level > 50 mg/L, level of ferritin > 500 ng/mL, level of d-dimers > 1000 ng/mL or level of lactate dehydrogenase > 250 U/L. Furthermore, patients were excluded if they were receiving supplemental oxygen at a rate greater than 10 liters per minute. The study entry criteria did not guarantee that tocilizumab was administered to patients with dyspnea or CRS markers. This raises doubts as to whether tocilizumab was administered in a timely manner, which could reduce its effectiveness.

Additionally, in a study by Salvarani et al., tocilizumab was considered ineffective in terms of preventing disease progression or clinical worsening compared to standard care. It is worth noting that their study had several limitations. The main disadvantages of their work were the failure to reach 80% of the power as the sample size was much smaller than calculated (126 vs. 398), as well as the selection bias of excluding patients with more severe disease, making it impossible to analyze the efficacy of tocilizumab in severe cases of COVID-19 pneumonia [19].

Our study had several limitations that we know of. They were related to its observational and retrospective nature. Immunomodulating treatment was selected by physicians on the basis of knowledge and guidelines, and was periodically limited by the availability of drugs and lack of randomization. We did not conduct a safety analysis of the drugs used due to the lack of access to such data. Electronic data recording may result in possible data entry errors, but we took every precaution to minimize such risks.

The main advantage of the current analysis was the collection of data from one center, which excluded the possibility of error related to different standards of individual tests. Another advantage of our analysis was the accurate assessment of lung involvement and its impact on the treatment process.

## 5. Conclusions

Recent research has elucidated the role of IL-6 blockade in the treatment of COVID-19. Recently, the REMAP-CAP [24] and RECOVERY [23] studies have provided evidence of the improved survival benefit of tocilizumab in patients with COVID-19-pneumonia. A large meta-analysis also found reductions in all-cause mortality associated with administration of IL-6 antagonists [27]. Our results suggest that the combination of tocilizumab with standard therapy (based on dexamethasone) provides a significantly better survival effect than standard therapy alone in patients with severe or critical lung involvement of COIVD-19 who have developed a cytokine storm characterized by high interleukin-6 levels (>100 pg/mL). Cytokine release syndrome has become a significant clinical problem in recent years. While tocilizumab may be effective in treating CRS, we still lack effective alternative treatments. Only further research into understanding the essence of CRS and possible immunomodulatory therapies can help to solve this problem.

## Figures and Tables

**Figure 1 jcm-11-02286-f001:**
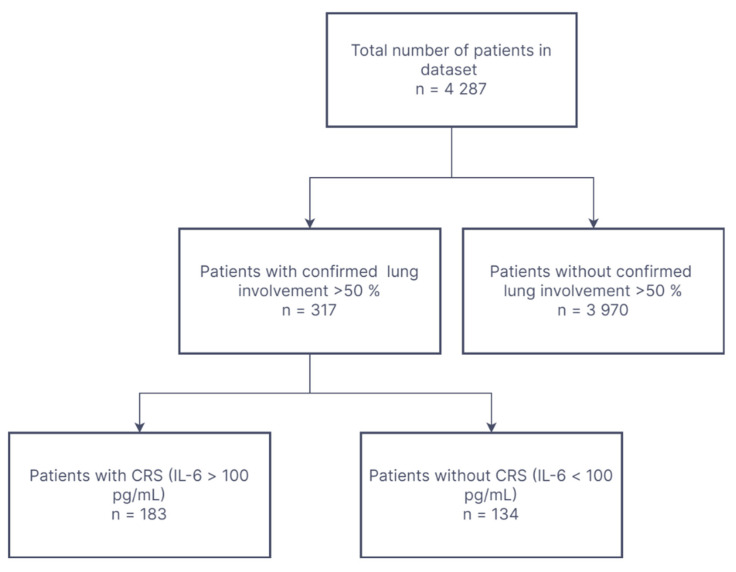
Flowchart of the study inclusion criteria.

**Figure 2 jcm-11-02286-f002:**
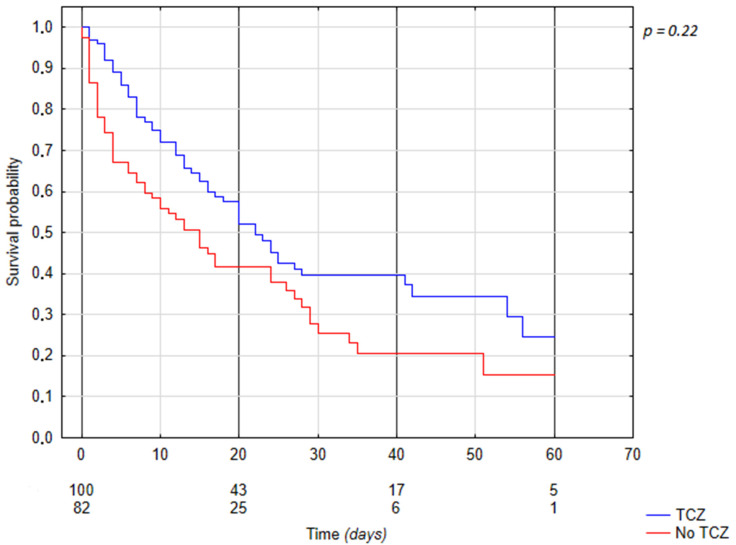
Kaplan–Meier curves displaying the estimated survival probability for TCZ-related treatment of patients with severe or critical COVID-19-related pneumonia.

**Table 1 jcm-11-02286-t001:** Baseline characteristics of all patients included in the study.

	Baseline Characteristics of All Patients Included in the Study			
	*n*	Median	Lower Quartile	Upper Quartile
Age, years	182	68.50	61.00	76.00
Percentage of Lung Involvement, %	182	59.61	54.52	67.38
WBC, ×10^3^/μL	181	9.05	6.37	11.88
NEU, ×10^3^/μL	181	7.10	5.20	9.90
LYM, ×10^3^/μL	181	0.80	0.50	1.10
RBC, ×10^6^/μL	181	4.57	4.15	4.91
HGB g/dL	181	13.60	12.30	14.70
HCT, %	181	39.50	36.10	42.70
Platelets, ×10^3^/μL	181	231.00	183.00	284.00
Procalcitonin, ng/mL	179	0.35	0.17	0.65
CRP, mg/L	182	183.97	133.75	258.28
IL-6, pg/mL	182	177.50	129.00	287.00
LDH, U/L	176	630.00	521.50	783.50
D-dimer, μg/L	178	1195.50	631.00	4387.00
Creatinine, mg/dL	182	1.13	0.87	1.48
AST, U/L	176	59.00	47.00	82.00
ALT, U/L	176	39.00	26.50	60.50
Troponin T, ng/L	174	26.80	15.70	49.00
GGTP, U/L	167	63.00	36.00	127.00
Bilirubin total, mg/dL	167	0.59	0.42	0.77

**Table 2 jcm-11-02286-t002:** Characteristics of groups admitted to ICU vs. not admitted to ICU.

Characteristics of Groups Admitted to ICU vs. Not Admitted to ICU				Cox Proportional Hazards Model for ICU Admission			
	ICU *n* = 59	Non-ICU *n* = 123	*p*	*p* Value	Hazard Ratio (HR)	Lower 95%CI HR Value	Upper 95%CI HR Value
Percentage of Lung Involvement, median (IQR)	59.06 (54.39–63.76)	63.51 (54.61–71.39)	0.049	0.161	1024	0.991	1058
RBC, ×10^6^/μL, median (IQR)	4475 (4.04–4.86)	4.7 (4.33–5)	0.022	0.441	1211	0.744	1974
CRP, mg/L, median (IQR)	175.2 (125.4–249.81)	205.71 (156.79–277.91)	0.021	0.004	1005	1002	1009
D-dimers, μg/L, median (IQR)	1488 (660–6501)	868 (594–1923)	0.009	0.158	1000	1000	1000
Troponin T, ng/L, median (IQR)	30.65 (18.7–54.25)	21.2 (12.8–39.1)	0.034	0.321	1002	0.998	1006
Age Group 18–65 (reference) vs. 65+	34 (58%)	33 (27%)	0.001	0.517	1236	0.651	2345
Hypertension n (%) Yes (reference)	38 (60%)	25(40%)	Chi-square Pearson *p* = 0.001	0.468	0.780	0.398	1526
Diabetes, n (%) Yes (reference)	26 (68%)	12 (32%)	Chi-square Pearson *p* = 0.001	0.021	2117	1119	4002
Remdesivir, n (%) Yes (reference)	28 (47%)	29 (24%)	Chi-square Pearson *p* = 0.001	0.645	1148	0.637	2070
Tocilizumab, n (%) Yes (reference)	29 (49%)	71 (58%)	Chi-square Pearson *p* = 0.277	0.064	0.575	0.320	1032

**Table 3 jcm-11-02286-t003:** Characteristic of surviving group compared to the patients who died.

	Characteristics of Surviving Group Compared to the Patients Who Died			Cox Proportional Hazards Model for Mortality			
	Survived *n* = 67	Died *n* = 115	*p* Value	*p* Value	Hazard Ratio (HR)	Lower 95%CI HR Value	Upper 95%CI HR Value
Percentage of Lung Involvement, median (IQR)	56.73 (53.71–63.76)	61.11 (54.78–68.34)	0.051	0.199	1017	0.992	1044
LYM, ×10^3^/μL, median (IQR)	0.9 (0.6–1.3)	0.7 (0.5–1.1)	0.024	0.438	0.854	0.574	1273
RBC, ×10^6^/μL, median (IQR)	4.78 (4.25–5.12)	4.46 (4.05–4.82)	0.004	0.797	0.891	0.368	2157
HGB g/dL, median (IQR)	14 (12.8–14.8)	13.4 (12–14.4)	0.016	0.867	0.957	0.570	1607
HCT, %, median (IQR)	41.2 (37.5–43.3)	39.2 (35.4–41.8)	0.026	0.826	1024	0.834	1256
ALT, U/L, median (IQR)	44 (36–69)	33.5 (23–55)	0.001	0.638	0.999	0.992	1006
Troponin T, ng/L, median (IQR)	19.1 (11.8–27.1)	34.55 (21.5–60.4)	0.000	0.013	1002	1001	1004
GGTP, U/L, median (IQR)	76.5 (45.5–158.5)	47 (31–95)	0.001	0.092	0.998	0.995	1001
Age Group 18–65 (reference) vs. 65+	35 (52%)	32 (28%)	Chi-square Pearson *p* = 0.001	0.003	0.466	0.286	0.761
Gender, n (%) Male(reference)	52 (78%)	72 (63%)	Chi-square Pearson *p* = 0.036	0.966	1012	0.611	1676
Procalcitonin, n (%) <2.0 ng/mL (reference)	65 (98%)	100 (88%)	Chi-square Pearson *p* = 0.016	0.683	0.840	0.364	1940
Creatinine Level, n (%) <2.0 mg/dL (reference)	66 (99%)	97 (84%)	Chi-square Pearson *p* = 0.002	0.048	0.499	0.251	0.993
Diabetes, n (%) Yes	11 (16%)	27 (23%)	Chi-square Pearson *p* = 0.258	0.346	0.789	0.482	1292
Tocilizumab, n (%) Yes (reference)	42 (63%)	58 (50%)	Chi-square Pearson *p* = 0.109	0.032	0.615	0.394	0.956

## Data Availability

The original anonymous dataset is available on request from the corresponding author at daniel.chober@pum.edu.pl.

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
