# Peer review of "Effectiveness of Tocilizumab in Patients with Severe or Critical Lung Involvement in COVID-19: A Retrospective Study"

_jcm, 2022, doi:10.3390/jcm11092286_

Round 1

Reviewer 1 Report

Please find above my recommendations following the STROBE statement checklist for cohort studies.

Abstract:

It would be helpful for readers to know when (period of time) patients were selected, as COVID-19 treatment has been changing from the first outbreak in 2020.

Methods

- Presenting key elements of the study design early in the paper is recommended. However, the key word “retrospectively” or “retrospective” are not mentioned in the methods section. So, please provide this information in this section.

- Main objective is stated but it is recommended to declare secondary outcomes as well (risk factors for ICU admission, etc..).

- It is also recommended to avoid giving results in the section methods. In the methods section the number of eligible patients are given before declaring the eligibility criteria. Please, provide this number in the results section as a result of the eligibility criteria. Moreover, please report in the results section the numbers of individuals at each stage of study (see results section).

- Defining and giving information about exposure, predictors, potential confounders, and effect modifiers is very important to identify participants, variables and avoid potential bias. In this line, it would be helpful for readers to provide the following information:

  • Give a clear eligibility criteria list for participants
  • Provide timings of each eligibility criteria: time to first dose of tocilizumab administered, time to chest computed tomography (C-CT); time to IL-6 level; etc.
  • Since comorbidities might play an important role in the main objective; comorbidities of the participants should be described to control potential bias. If not, please give reasons.
  • Since treatment recommendations have been changed from March 2020 to January 2022; treatments of the participants included should be given as well as the supportive oxygen supplementation therapy. Moreover, in the reference 21 (Flisiak R, Horban A, Jaroszewicz J, et al. Management of SARS-CoV-2 infection: recommendations of the Polish Association of 363 Epidemiologists and Infectiologists as of March 31, 2020. Polish Arch Intern Med. 2020;130(4):352-357. 364 doi:10.20452/PAMW.15270), corticoids were recommended when tocilizumab is used (“Glucocorticoids in case of deterioration of respiratory function (necessary especially when tocilizumab is used”). As the use of corticoids has shown to be beneficial for COVID-19 patients; the use of corticoids should be described and included in the methods section.
  • In the same way, patients receiving other potential drugs against COVID-19 should be identified.
  • Please, provide information about who decide the administration of tocilizumab; if there was a common criterion, or the final decision to use tocilizumab was at the discretion of the treating clinician.

- Ethical issues. Since this is a retrospective study and tocilizumab was used as an off-label drug; please clarify if the informed consent was only for the use of tocilizumab as an off-label drug.

- Statistics. Please provide enough information about statistical methods, especially about the multivariate proportional Cox hazards models mentioned in the abstract. There is no information about these models or methods to build these models. In this line, it is recommended to describe the statistical methods to control for confounding such as multivariate regression; and describe adjusted methods (Backward, forward methods, Akaike’s Information Criterion, etc...). In this section, explain which variables would be used to adjust and the method used as well as the reasons to include these variables. Therefore, reasons and methods to build the final model in a multivariate regression should be given.

Results

- As mentioned before, please provide information about numbers of individuals at each stage; eg numbers potentially eligible, reasons for no eligibility...

-  Please give characteristics of the participants, including treatment regimen (especially corticoids and other drugs with potential effect against COVID-19); and comorbidities; as both variables could be potential cofounders.

- Table 1 and Table 2 show duplicate data, please unify. Besides, there are items with low value such as WBC, NEU, LYM, RBC,... and avoid duplicates such as Male and Female; Admitted ICU and Not admitted; Survived... also in Table 3 and Table 4.

- It is recommended do not provide p values when 95% confidence intervals are given for HR and OR.

- Table 3 and Table 4 show also duplicate data, please unify. In addition, comorbidities should be. Identically, please unify Table 5 and 6.

- Cox proportional-hazard models given in the results section should be reviewed since all the variables have been included in the final model. The total number of participants is very short (182) to include such as high number of variables in the final model (>20). In general, it is recommended to have at least 10 participants for each variable included in the regression model.

Discussion

-  As mentioned before, the use of corticoids should be discussed in this cohort, since effectiveness of tocilizumab has been related to the use of corticoids.

- The results have shown a high mortality, even in not ICU patients (70 patients of 123 non ICU patients, 56.9%). Please tackle this point regarding the reasons for these results, as well reasons for not admission to ICU (eg no availability of ICU beds, reasons or criteria for ICU admission, etc...).

Author Response

Dear Reviewer,

we would like to very warmly thank the reviewer for the diligent and the most valuable comments. We have taken every effort to include them accurately in the manuscript, and hope that the manuscript is suitable for further processing and publication in JCM.

Below please find responses to the specific comments, also highlighted in the text.

Reviewer 2 Report

Thank you for inviting me as reviewer to read and comment on paper by Daniel Chober et al entitled “Effectiveness of Tocilizumab in Patients with Severe or Critical lung involvement in COVID-19: A Retrospective Study” submitted to the JCM. The title seems nice and suitable for this type of paper, but I have some extra comments suggestive for the paper. The writing skill is acceptable and my main comments are targeting the scientific content of the paper. Once more, the paper is well-written so my comments are mostly complementary to increase the readability of the content.

  • This sentence “Efficacy of Tocilizumab in patients with severe or critical lung involvement with Covid-19-related pneumonia is unclear, therefore we aimed to bridge this gap in knowledge.” Is the main rationale behind this article? I am not agree with authors because of the efficacy of Tocilizumab for these patients are not completely unclear since currently many of clinics are using it for patients with severe prognosis. Thus, this aim and any related claims should be discarded in this revised manuscript.
  • Importance of usage of tocilizumab is not mentioned in this paper, why? I feel this should be mentioned in the introduction otherwise this lacks.
  • Nomination of such patients with indication of IL-6 more than 100 pg/ml as target subjects should be correctly referenced, now I CAN not see the suitable reference for this clinical terminology.
  • Page 1 line 33-34, this is not correctly referenced. Please add a new and exact research that support this claim (I mean the mortality rate for severe patients).
  • Page 2, line 66, and please remove this wording about the guideline! At least move it to the discussion, i have some suggestions to enrich your discussion as well.
  • Gathering data for 4278 patients is not very clear, this is main lacking in methods section. How long it lasts until you collect their data? When they admitted to the clinical center? Or centers? These type of data should be clearly stated.
  • The baseline data for these 182 patients is nicely mentioned in table 1, I appreciate it.
  • Basically, I am wondering how the ethics committee approved the permission to call 45% of patients as control without receiving the TCZ. This made me worry! Please clarify it.
  • I can imagine that the authors are presenting whole data they retrieved from this analysis but honestly the table 2 is nonsense! What significant data is presenting there?
  • If possible I would recommend to merge table 3 and 4, indeed , we are not watching high amount of significant data out of them, please recheck them, however, it ups to the authors, this is not mandatory.
  • Page 10, line 210, start the discussion should be with more general idea about the COVID-19 patients and their clinical management. Recently WHO call for a formulation of using Tocilizumab in treating the COVID patients, so I would recommend the authors to add this update reference it somewhere here. Application of TCZ is useful while the WHO recommendation is available otherwise any result by this authors are foggy!
  • Page 11, line 265, please remove this claim, this make your article almost equal to zero!
  • Lastly, please make a great literature review only for 2022 papers indexed in the databases, you should have review them and rewrite your conclusion in this light. Covid pandemic is an evolving phenomena so any ideas should considered an updated picture of the situation otherwise no one listen it!

Author Response

(The authors gave the same response as above.)

Round 2

Reviewer 1 Report

No comments. Thank you for considering the comments. Changes have been made in the paper in order to fit the suggestions given.